# Pharmacogenomics of Anti-TNF Treatment Response Marks a New Era of Tailored Rheumatoid Arthritis Therapy

**DOI:** 10.3390/ijms23042366

**Published:** 2022-02-21

**Authors:** Tomasz Wysocki, Agnieszka Paradowska-Gorycka

**Affiliations:** Department of Molecular Biology, National Institute of Geriatrics, Rheumatology and Rehabilitation, Spartańska 1, 02-637 Warsaw, Poland; agnieszka.paradowska-gorycka@spartanska.pl

**Keywords:** rheumatoid arthritis, pharmacogenomics, anti-TNF treatment

## Abstract

Rheumatoid arthritis (RA) is the most commonly occurring chronic inflammatory arthritis, the exact mechanism of which is not fully understood. Tumor Necrosis Factor (TNF)-targeting drugs has been shown to exert high effectiveness for RA, which indicates the key importance of this cytokine in this disease. Nevertheless, the response to TNF inhibitors varies, and approximately one third of RA patients are non-responders, which is explained by the influence of genetic factors. Knowledge in the field of pharmacogenomics of anti-TNF drugs is growing, but has not been applied in the clinical practice so far. Different genome-wide association studies identified a few single nucleotide polymorphisms associated with anti-TNF treatment response, which largely map genes involved in T cell function. Studies of the gene expression profile of RA patients have also indicated specific gene signatures that may be useful to develop novel prognostic tools. In this article, we discuss the significance of TNF in RA and present the current knowledge in pharmacogenomics related to anti-TNF treatment response.

## 1. Introduction

RA is a complex and heterogenous disease; what differentiates patients with RA are the number of affected joints, antibodies and serum cytokines levels, and the severity of joint destruction [1]. TNF (tumor necrosis factor) is a proinflammatory cytokine which exerts multiple biological activities in RA. Blocking TNF with antibodies became a standard of modern RA therapy in patients not responding to conventional synthetic disease-modifying antirheumatic drugs (csDMARDs), e.g., methotrexate. TNF inhibitors include infliximab, adalimumab, etanercept, golimumab, and certolizumab pegol, each having a similar efficacy in RA. TNF inhibitors remain the most practically useful biological drugs in RA, but their effectiveness is diverse in individual cases, which reflects the heterogeneity of RA. To date, there are no reliable tools to predict individual patient response to TNF inhibitors. The diversity of treatment options has opened new avenues for the development of precision therapy for RA, which aims to tailor the specific treatment for patients based on their individual characteristics. Pharmacogenomics is a key part of precision medicine, which studies the relationship between genomic variations and their effect on drugs. Currently, pharmacogenomic testing is used to predict clinical outcomes, e.g., CYP2C19 in patients taking clopidogrel, T315I mutation, which confers resistance to BCR-ABL inhibitors (imatinib, dasatinib, and nilotinib) in chronic myeloid leukemia, and ERBB2/HER2, which influences the resistance to trastuzumab. In oncology, genomic platforms, based on multigene RNA expression profiles to predict the response to therapies, have also been developed, e.g., PAM50 [2]. Pharmacogenomic markers may predict an increased risk of severe drug toxicity, e.g., the CYP2D6*10 allele in tamoxifen therapy of breast cancer. In rheumatology, a few reports have shown that common variants in genes that encode NAT2 and ABCG2 are associated with toxicity and response to sulfasalazine, respectively. Two main techniques, such as immunohistochemistry (IHC) and fluorescence in situ hybridization (FISH), estimating protein expression and gene amplification, respectively, are already used in clinical practice. Current advances in pharmacogenomics may enable the optimization of RA therapy by reducing the time to achieve remission. In addition, it may help to limit some adverse effects of anti-TNF agents, which include serious infections, malignancies, congestive heart failure, drug-induced lupus, and demyelinating disorders. Identifying patients with predicted better or worse response to TNF inhibitors would also facilitate the improving of the cost-effectiveness of RA treatment.

## 2. The Pivotal Importance of TNF in RA

TNF is a pleiotropic proinflammatory cytokine which is produced mainly by activated monocytes and macrophages and, to a lesser extent, by T-lymphocytes. The T cell-derived response is thought to be of particular importance in triggering TNF efflux by synovial macrophages [3]. Citrullination of proteins at sites of inflammation is a key process leading to the activation of macrophages in RA. Citrulline-specific T cells display predominately Th1 and Th17 phenotypes, and their effector cytokines are interferon gamma (IFN-γ) and interleukin (IL)-17, which activate macrophages. Macrophages can be also activated by direct contact with T cells [4]. Anti-citrullinated protein antibodies (ACPAs) are implicated in immune complex formation. Sokolove et al. demonstrated that ACPA-IC is an inducer of TNF production by triggering the innate immune receptor Toll-like receptor (TLR) 4- and the crystallizable fragment gamma receptor (FcγR)-dependent signaling in macrophages [5]. In addition, ACPAs selectively activate extracellular signal-regulated kinase 1/2 (ERK1/2) and c-jun N-terminal kinase (JNK), facilitating nuclear factor κB (NFκB) pathways and TNF production [6]. ACPAs can also suppress the expression of microRNA let 7a, which also contribute to enhancing TNF expression [7]. TNF exists in transmembrane and soluble forms which have different biological functions. The soluble TNF (sTNF) is cleaved from transmembrane TNF (tmTNF) by metalloprotease TNF-alpha-converting enzyme (TACE). The effects of TNF are mediated through two distinct receptors: TNF receptor 1 (TNFR1), expressed ubiquitously, and TNFR2, expressed primarily in immune cells, neurons and endothelial cells. TNFR1 is activated by both sTNF and tmTNF, whereas TNFR2 has a high affinity to only membrane-bound forms [8]. Stimulation of TNFR1/2 exerts distinct cellular responses. TNF signaling through both TNFR 1 and TNFR2 leads to the expression of NF-κB and activator protein 1 (AP-1) target genes associated with cell survival. Activation of TNFR1 is thought to primarily induce proinflammatory responses, whereas TNFR2 mostly mediates local homeostatic signals. TNFR1 is also capable of inducing cell death responses, including apoptosis (programmed cell death) and necroptosis (uncontrolled cell death). Apoptosis, mediated by the caspase-8 activation pathway, is thought to induce paradoxical anti-inflammatory and immunosuppressive responses. In contrast, necroptosis triggers local inflammation [9]. The factors determining whether the pathway leading to cell survival, apoptosis or necrosis is activated, is not well understood. Nevertheless, the ubiquitination status of receptor-interacting serine/threonine protein kinase 1 (RIPK1) seems to be crucial. The ubiquitylated RIPK1 leads to pro-survival signaling, whereas non-ubiquitylated RIPK1 induces either apoptosis or necroptosis. The long form of the FLICE-inhibitory protein, also known as c-FLIP(L), is another important regulator of both apoptosis and necrosis pathways. c-FLIP(L) is a major apoptosis inhibitor, similar in structure to caspase-8, which is also capable of forming a proteolytically active complex with caspase-8, which cleaves RIPK1 and RIPK3, and blocks RIPK1/RIPK3/MLKL-dependent necroptosis [10].

## 3. Diverse Influence of TNF on Target Cells in RA

### 3.1. Fibroblast-like Synoviocytes

TNF is an extraordinarily pleiotropic cytokine influencing different types of cells (Figure 1). Synovium is the principal site of immune disorders and inflammation in RA, and consists of the continuous surface layer of cells (intima) and the underlying tissue (subintima). The healthy synovial membrane contains relatively few cells and is predominantly composed of two types of synoviocytes, forming one to two cells per layer: type A are macrophage-like synoviocytes (MLS), originating in the bone marrow, capable of phagocytosis, whereas type B, also known as fibroblast-like synoviocytes (FLS), are responsible for synthesis of the synovial fluid hyaluronan [11]. In healthy synovium, type A and type B synoviocytes exist in relatively equal proportions. The extension of synoviocytes in RA form the locally invasive synovial tissue, called the pannus [12]. The synovial lining is greatly hypertrophied, and both types of synoviocytes proliferate, but MLS are more abundant [13]. The role of MLS in RA is not entirely clear, but presumably MLS induce invasiveness of FLS via the secretion of proinflammatory cytokines, including TNF, which mostly contribute to joint destruction.

TNF-mediated signaling in FLS is presumably crucial for arthritis initiation [14,15]. FLS are a major source of multiple proinflammatory cytokines and chemokines, which lead to recruitment of more immune cells. At the transcriptional level, TNF in RA regulates the expression of proinflammatory genes in a cell type-specific manner. The TNF-inducible proinflammatory genes exhibit unremitting expression in FLS. The sustained expression in FLS is presumably a result of an alteration of the chromatin structure in regulatory elements of the genes, resulting in the continuous upstream activation of NF-κB and MAPK signaling.

### 3.2. Macrophages

In macrophages, in contrast to FLS, the proinflammatory genes are only transiently expressed in response to TNF [16]. Moreover, on a prolonged exposure to TNF, macrophages display tolerance induction [17]. However, such desensitization may be blocked by type I interferons. TNF and type I IFNs presumably cooperatively regulate chromatin accessibility at inflammatory gene promoters, facilitating robust transcriptional responses to weak signals [18]. In addition, in macrophages and endothelial cells, TNF induce the IFN-β autocrine loop via activation of interferon regulatory factor 1 (IRF1) and interferon regulatory factor 3 (IRF3) [19]. Treatment with anti-TNF antibodies strongly modulate functions of macrophages in inflammatory conditions. The process is IL-10/STAT3 pathway–dependent. Upon anti-TNF blocking, the surface expression of pro-inflammatory markers (CD40 and CD60) is downregulated, whereas expression of phagocytic receptors (CD16, CD163, MER proto-oncogene tyrosine kinase) is increased. In addition, anti-TNF agents also suppress production of pro-inflammatory cytokines (IL-6, IL-12, TNF) in macrophages [16].

### 3.3. Bone Marrow Stromal Cells and Osteoclastogenesis

RA-associated bone loss is a result of the disturbance between bone formation and bone resorption. Proinflammatory cytokines, including TNF, are essential for osteoclastogenesis. TNF stimulate the expression of receptor activator of NF-κB ligand (RANKL) by bone marrow stromal cells, mainly fibroblasts and activated synovial T cells. RANKL is crucial for propagating osteoclasts maturation. TNF can also stimulate osteoclast precursors directly through TNFR1 signalling. The soluble TNF is responsible for mobilizing osteoclasts from the bone marrow [20,21]. In addition, TNF is a potent inhibitor of osteoblast differentiation [22,23]. In contrast, TNFR2-dependent signalling inhibits osteoclastogenesis [24].

### 3.4. Neutrophils

In RA, TNF exposure on neutrophils up-regulates several genes involved in the NF-κB signaling pathway, particularly TNF itself. In addition, TNF induces expression of anti-apoptotic genes, e.g., *TNFAIP3 (A20), BCL2A1, CFLAR (FLIP)*, and *FAS*, while expression of some pro-apoptotic genes, including *APAF1, CASP8, CASP10, FADD, TNFRSF1A* and *TNFRSF1B*, is down-regulated. These transcriptional changes allow neutrophils to delay apoptosis and thereby enhance inflammation. Interestingly, these effects of TNF on human neutrophils seem to be concentration-dependent, since high concentrations may paradoxically promote apoptosis. During continuous exposure to TNF, desensitization of TNF-inducible expression in neutrophiles occurs, which is presumably a result of a down-modulation of both TNFR1 and TNFR2 receptors [25].

### 3.5. T Cells

The relationship between TNF and T cell responses is complex, since T cells both respond to and produce TNF. TNF is of a key importance for proper T cells development. It is responsible for modulation of T cells development in thymus, as well as generation of efficient adaptive immune responses in lymph nodes. TNF is necessary for the activation of effector, memory, and regulatory T cells. Extensive production of TNF influence maturation of the antigen-presenting dendritic cells, and thus recalls efficient activation of naïve T cells. In addition, TNF was experimentally shown to promote leukocyte adhesion and diapedesis, which enable T cells invading the inflamed tissue [26].

Expansion of Th17 cell subset is hypothesized to be an important mechanism of peripheral tolerance breakdown in RA. The increase in Th17 was reported in RA peripheral blood, synovial fluid and tissue, however Th17 lymphocytes are rarely found at inflammatory sites in comparison with other T cell subsets. This is presumably due to the rapid transition of Th17 lymphocytes to the non-classic Th1 phenotype, induced by TNF. Shifted Th17 cells become particularly aggressive and presumably are of particular importance in pathogenesis of inflammatory disease, including RA [27].

Much knowledge about the influence of TNF on lymphocytes in RA is provided also by data from studies evaluating the effect of treatment with anti-TNF drugs. Interestingly, in long-term observation of anti-TNF-treated patients, the number of circulating Th1, Th2 and Th17 lymphocytes was reported to increase [28,29]. Non-responders are also associated also with higher baseline Th17 frequencies, what may be a result of the impaired peripheral tolerance and paradoxical activation of Th1 and Th17 responses in this group of RA patients [30,31].

Dulic et al. also showed that, as compared with non-responders, anti-TNF responders in long-term observation were characterized with lower proportions of total CD4+ T cells and higher proportion of late activation CD4+HLA-DR+ T cells. In the same study, T cells expressing the early activation marker CD69 were indicated as promising predictive marker of anti-TNF blocker response [28].

TNF appears to inhibit the suppressive functions of Tregs, which are essential in preventing autoimmunity. Tregs prevent optimal costimulation by antigen presenting cells, and further response related to cytotoxic T cells [32]. The mechanism of such TNF-mediated inhibition is downmodulating the expression of the key transcription factor FoxP3. In line with the above, the anti-TNF treatment has a potential to restore the suppressive function in Tregs. The ratio of Th17/Treg in anti-TNF-treated patients, regardless of the effectiveness of therapy, tends to approach the ratio observed in healthy controls [28,33]. Under certain conditions, TNF may also increase Tregs differentiation and promote immunosuppressive effects, which appears to be TNFR2-dependent. The expression of TNFR2 on Tregs is decreased in autoimmune diseases, limiting the importance of this effect in RA. Based on these findings, the development of new anti-TNF therapies should be focused on a more selective blockade of TNFR1-dependent signaling [32].

The TNF inhibitor golimumab was shown to largely affect peripheral memory T cell responses. In long-term observation, patients treated with golimumab displayed increased effector memory T cells and antigen-specific T cell cytokine production [34].

As for B cells, conflicting results have been reported. TNF blocking with infliximab was found to increase the frequencies of circulating memory B-cells, whereas another study with etanercept showed the opposite effect [35].

### 3.6. Effects of TNF on Angiogenesis

Formation of new capillaries from existing vessels, termed angiogenesis, is a key event in the formation and maintenance of pannus [36]. Proinflammatory cytokines, including TNF, stimulate synovial fibroblasts to release vascular endothelial growth factor (VEGF). VEGF stimulates endothelial cell proliferation, and thus angiogenesis. TNF also affects the later stage of angiogenesis by the increased production of angiopoietin 1, important for newly-formed blood vessel stability [37].

### 3.7. Influence of TNF on Endothelial Dysfunction

TNF contributes to the increased risk of cardiovascular comorbidities in RA patients. Vascular injury is a result of the disruption of the endothelial barrier. TNF via TNFR1 signaling destabilizes the endothelial skeleton by actin rearrangement and microtubule destabilization [38]. In addition, TNF was shown to induce degradation of the endothelial glycocalyx [39]. TNF also impairs nitric oxide formation, which is of key significance for endothelium-dependent vasodilatation. TNF-signalling decreases NO generation by TNFR1-dependent inhibition of endothelial NO synthase (eNOS) expression and accumulation of the endogenous eNOS inhibitor, but also influences the enhanced removal of NO. Moreover, TNF stimulates production of vascular reactive oxygen species (ROS) by upregulated expression of NADPH-dependent oxidase (NOX) subunits. Upon TNF activation, ROS contribute to TNF-induced NF-kB activation, decreased NO bioavailability and progression of atherosclerosis [40].

## 4. TNF Inhibitors Characteristics

According to a recent 2019 update of the European League Against Rheumatism (EULAR) recommendations for the management of rheumatoid arthritis, TNF inhibitors should be applied in patients with insufficient response to an initial course of conventional synthetic (cs) DMARDs (including methotrexate, sulfasalazine, leflunomide) and the presence of poor prognostic factors [41].

Each of the TNF inhibitors show substantial structural differences (Figure 2). TNF inhibitors include monoclonal antibodies, e.g., infliximab (IFX), adalimumab (ADA), golimumab (GOL) and certolizumab pegol (CZP), and circulating receptor fusion proteins such as etanercept (ETN).

ADA is a fully human monoclonal IgG1 antibody which binds specifically to TNF, and inhibits its activity by blocking its binding to the p55 and p75 TNF receptors on the cell surface. ADA also influences a biological response induced or regulated by TNF, including changes in the concentration of intercellular adhesion molecules responsible for leukocyte migration (ELAM-1, VCAM-1 and ICAM-1). In RA patients, a rapid decrease in the levels of acute phase reactants (C-reactive protein, erythrocyte sedimentation rate [ESR]), cytokines (IL-6) and matrix metalloproteinases (MMP-1, MMP-3) is observed. ADA is administered subcutaneously (s.c.) at a dose of 40 mg every two weeks. After administering s.c., the drug is absorbed and distributed slowly, with t_max_ of about five days, and mean bioavailability of 64%. The elimination half-life (t_1/2_) is approximately 14 days [42,43].

GOL is a fully human monoclonal IgG1 antibody that forms high affinity complexes with both sTNF and tmTNF, preventing TNF from binding to its receptors. Similarly to ADA, the binding of human TNF by golimumab neutralizes TNF-induced cell surface expression of the E-selectin adhesion molecule, the vascular intercellular adhesion molecule (VCAM-1), and the intercellular adhesion molecule (ICAM-1) of endothelial cells. In vitro, golimumab inhibits the secretion of IL-6, IL-8 and TNF-induced granulocyte colony stimulating factor (GM-CSF). In patients receiving golimumab, a decrease in C-reactive protein levels is observed, resulting in significant reductions in IL-6, ICAM-1 molecules, MMP-3) and vascular endothelial cell growth factor (VEGF). GOL is administered either s.c. at a dose of 50 mg every month, or intravenously (i.v.). After administering s.c., the mean t_max_ is two to six days and the mean absolute bioavailability is 51%. t_1/2_ is approximately 9 days [43,44].

IFX is a chimeric monoclonal antibody (mAb) composed of a murine variable region and a human IgG1 constant region, which has high affinity for both sTNF and tmTNF. In vivo, IFX rapidly forms stable complexes with human TNF, resulting in the loss of biological activity by TNF, and a decrease in the concentration of IL-6 and C-reactive protein. In RA patients, the standard dose of IFX is 3 mg/kg i.v. at weeks 0, 2, and 6 (induction therapy), followed by maintenance therapy every eight weeks. t_1/2_ ranges from 7.7 to 9.5 days [43,45].

Certolizumab pegol is a PEGylated Fab’ fragment of a humanized monoclonal antibody, obtained by expression in E. coli cells and conjugation with polyethylene glycol (PEG). PEG conjugation prolongs the t_1/2_ to values comparable to the t_1/2_ of a complete antibody. CZP has a high affinity for both sTNF and tmTNF. CZP does not contain the fragment crystallizable region (Fc) that is present in complete antibodies, therefore it does not bind complement, and does not cause antibody dependent cellular cytotoxicity. In addition, it does not induce apoptosis in vitro in human monocytes or peripheral blood lymphocytes, nor does it induce neutrophil degranulation. The recommended starting dose of CZP for adult patients is 400 mg at weeks 0, 2 and 4, followed by a maintenance regimen every four weeks. Bioavailability is approximately 80%, and t_1/2_ is approximately 14 days [43,46].

ETA is a human fusion protein composed of the extracellular domain of TNF receptor (TNFR2/p75) and the Fc region of IgG1. Soluble receptors in dimeric form, such as ETA, have a greater affinity for TNF than monomeric receptors and are therefore much more potent competitive inhibitors of TNF. The presence of an immunoglobulin Fc fragment as a binding element prolongs serum t_1/2_, which is approximately 70 h. ETN works by competitive inhibition of TNF to prevent attaching to TNFR receptors, rendering TNF biologically inactive. The drug is administered s.c. at a standard dose of 50 mg every week. The estimated bioavailability is 76% [43,47].

## 5. Primary and Secondary Non-Responsiveness to TNF Inhibitors

It is estimated that around 25% of anti-TNF-treated RA patients fail to achieve initial treatment targets (primary non-response) [48,49]. If a patient is primary non-responder to one TNF inhibitor, it is not likely to respond on the other one, since all anti-TNF target the same molecular pathway [50]. Furthermore, approximately 55% of primary responders relapse within 12 months of treatment, which is secondary non-response [48,49]. The precise mechanism of such unresponsiveness to TNF blockers is not entirely clear. According to the 2019 update of the EULAR recommendations for the management of rheumatoid arthritis, if a patient treated with TNF inhibitor is a primary non-responder or develops secondary non-responsiveness over time, the treatment with this drug should be discontinuated. It is recommended to change it to another anti-TNF drug, either bDMARD from another class or a JAK-inhibitor [41].

## 6. Anti-TNF Pharmacogenomics

### 6.1. Genetic Loci

The genome wide association study (GWAS) approach is useful for identifying potentially causal single-nucleotide polymorphisms, which is predictive for an anti-TNF therapy outcome. An analysis of the large Dutch Rheumatoid Arthritis Monitoring (DREAM) registry indicated that eight genetic loci explain 3.8% of the variance in treatment response. Two single nucleotide polymorphisms (SNPs), rs12142623 and rs4651370, are located in close proximity to the PLA2G4A gene (Table 1). The protein encoded by PLA2G4A is a phospholipase enzyme involved in the generation of eicosanoids, which regulate inflammatory responses. The other important markers are rs2378945, located in the *nucleotide-binding protein-like (NUBPL)* gene, which is required for the assembly of the respiratory chain NADH dehydrogenase (complex I), and rs1813443, located in the intronic region of *contactin 5 (NCTN5)*, a member of the immunoglobulin superfamily. The other identified markers are rs4411591, rs7767069, rs1447722, and rs1568885, the biological roles of which remain unclear. Nevertheless, none of the identified eight SNP reached a genome-wide level of significance, thus their occurrence tends to be more prognostic than predictive [51]. The results of a recent meta-analysis of three large cohorts, including the DREAM registry, as well as the REPAIR consortium and the DANBIO registry, underlined the role of *LINC02549* rs7767069 and either copy of the *LARRC55* rs717117G allele. Carriers of the rs7767069 allele show an increase in the number of T cells expressing the memory markers CD45RO and CD45RA, as well as the increased serum levels of soluble scavenger CD5 and CD6 receptors.

Carriers of the *LARRC55* rs717117G allele show decreased levels of IL-6 after stimulation of PBMCs, which suggest the possible link between reduced IL6-mediated anti-inflammatory response and inadequate response to iTNF in this group of patients [52]. The genome-wide significant association with primary response to etanercept was found in the *MED15* gene (rs113878252), which is presumably responsible for polymerase II transcription. The other highly significant SNP in this study was in the *MAFB* locus (rs6065221), encoding the basic leucine zipper transcription factor that plays an important role in the regulation of lineage-specific hematopoiesis [53]. Two SNPs, rs6427528 and rs1503860, which are associated with higher *CD84* gene expression in peripheral blood mononuclear cells, were indicated as a potential predictors of response to etanercept [54]. In single studies, the rs1800629 polymorphism in the *TNF* gene, rs17301249 in the *EYA4* gene, rs1532269 mapping to the *PDZD2* gene, and rs2812378 in the *CCL21* gene, were also associated with responsiveness to TNF inhibitors [55,56,57].

### 6.2. Gene Expression

#### 6.2.1. Pathotypes of RA Synovium and Synovial Tissue Gene Expression

Synovial biomarkers may be useful tools to predict anti-TNF clinical response (Table 2). In RA, a marked cellular and molecular variety of the inflamed synovium tissue is observed [58]. The heterogeneity of RA synovitis is reflected in distinct cellular and molecular signature (pathotypes). At first, Dennis et al. distinguished four pathotypes: lymphoid, myeloid, pauci-immune (low inflammatory), and fibroid [59]. However, the relationship between these and clinical phenotypes is ambiguous. The analysis of a large cohort of early, treatment-naïve RA patients revealed the existence of three specific pathotypes: (1) lympho-myeloid, characterized by predominant lymphoid infiltrates (including T cells, B cells and plasma cells), (2) diffuse-myeloid, with predominant sublining macrophages and lacking B cells/plasma cells aggregates, and (3) pauci-immune-fibroid (low inflammatory), with hardly any immune cells and prevalent resident fibroblasts. Four molecular phenotypes of RA synovium correspond with specific synovial tissue gene expression. The myeloid pathotype has enhanced expression of nuclear factor kappa-light-chain-enhancer of activated B cells (NF-κB) pathway genes (including *TNF, IL-1β, IL-1RA, ICAM1*, and *MyD88*), the inflammatory chemokines (*CCL2, IL-8*), as well as neutrophiles, macrophage and osteoclast-associated genes (*S100A12, CD14, OSCAR*, respectively). The lympho-myeloid pathotype was associated with the highest expression of B cell- and plasmablast lineage genes (including *CD19, CD20, CD27, IGLL5, XBP1, immunoglobulin heavy and light chains, CD38 and CXCL13*). The fibroid pathotype is characterized by high expression of fibroblast growth factor genes (*FGF2, FGF9*), bone homeostasis–associated genes (*BMP6, TNFRSF11b/osteoprotogerin*) and the *Wnt* and *TGFβ* pathways genes. The pauci-immune pathotype has the most diverse expression profile and all of the above-mentioned patterns may be found, but the additional expression of the *SFRP3/FRZB* gene, involved in Wnt signaling and the regulation of bone development, may be elevated. All groups show increased expression of IL-6, the IL-6 receptor, and associated STAT3 protein transcription factor pathway genes [59,60].

In general, the lymphoid pathotype is linked to more aggressive disease, RF and ACPA seropositivity, and a higher proportion of patients requiring biological therapy [61,62]. Furthermore, patients with low expression of *sICAM1* and high expression of *CXCL13* corresponding to the lymphoid pathotype were shown to have a better clinical response to anti-TNF treatment as compared with myeloid pathotype-associated *sICAM1*-high/*CXCL13*-low patients. Interestingly, the opposite relationships have been found in IL-6R–treated patients [59]. In contrast, patients categorized as pauci-immune are associated with having a poor response to TNF blockers [63].

Aterido et al. performed the analysis of transcriptomic association between selected gene coexpression modules in the RA synovium and the clinical response to TNF inhibitors, including ADA and IFX. Two of the 13 analyzed modules, which were significantly enriched for genes involved in the nucleotide metabolism and epigenetic marks from CD4+ regulatory T cells, have been found to be significantly associated with the response to anti-TNF drugs [64].

#### 6.2.2. Whole-Blood Transcriptome

In general, the peripheral whole-blood transcriptome in RA patients is less variable than the synovial one. However, in RNA-seq analysis a substantial association between type I IFN response genes expression (*IFI27, ISG15, IFI44L, OASL, USP18, RSAD2*, and *LY6E*) in peripheral whole blood and the lympho-myeloid pathotype has been shown [60].

#### 6.2.3. Peripheral Blood Mononuclear Cells

Beyond the above, the link between gene expression patterns of peripheral blood mononuclear cells (PBMCs) and response to treatment have also been found. PBMCs include lymphocytes (T cells, B cells and NK cells), monocytes and dendritic cells. Póliska et al. have identified five genes whose baseline expression is upregulated in PBMCs in TNF-responders: (1) *HLADRB4* (encoding class II molecules, expressed in antigen-presenting cells, e.g., B lymphocytes, dendritic cells, macrophages), (2) *TMEM176A* and (3) *TMEM176B* (encoding transmembrane proteins, involved in the process of maturation of dendritic cells), (4) *IFI44* (interferon-related gene, negative modulator of IFN responses) [65], and (5) *PLSCR1* (encoding phospholipid scramblase 1, involved in negative regulation of FcR-mediated phagocytosis in macrophages) [66,67].

In peripheral blood monocytes of RA patients, differential IFN pathway activation prior to anti-TNF treatment has been observed in subsets of responders and non-responders. In addition, TNF inhibitor response groups, which were defined by type I IFN, were shown to correlate with the expression of *JAK1*, *IFI27 IFNAR1*, *IRF1*, *TNFA*, *TLR4*, *MYD88*, *CD86*, and *IL8* [68].

Only recently, Tao et al. reported building the novel machine learning model based on the analysis of transcriptome signatures from PBMCs, monocytes, CD4+ T cells, and methylation signatures from PBMCs, which enable prediction of anti-TNF response in patients with RA. In the study, gene expression and DNA methylation profiling was performed in two cohorts of patients, who were going to begin either ADA or ETN therapy. In both cohorts of patients, genes differentially expressed between responders and non-responders included those involved in DNA and nucleotide binding, such as RFX2, IRF8, and TAF1, TAF11, FOXO4. Additionally, in the ETN cohort TRAF6, representing the TNF receptor signaling gene, was also differentially expressed. The interesting issue was that, in both cohorts, a distinct hypermethylation pattern between responders and non-responders was observed, and this may be of particular importance to understanding the importance of epigenetics in non-responsiveness to TNF inhibitors. The tool presented by the authors is undoubtedly interesting; however, it is worth noting that the indicated model reached an overall accuracy of 85.9% in the ADA cohort and 79% in the ETN cohort, which may represent insufficient levels of reliability for use in clinical practice [69]. Therefore, it is very important to continue the search for further markers, the identification of which would improve the existing model.

#### 6.2.4. Neutrophiles

The specific gene expression profiles in peripheral blood neutrophiles can also predict a subsequent response to TNF blockers. In a study by Wright et al., the type I IFN signalling was identified as the most enhanced pathway in anti-TNF responders. The expression levels of the 10 IFN-regulated genes *CMPK2, IFI44L, IFI6, IFIT1B, LY6E, OAS1, OAS2, OAS3, RSAD2,* and *USP18* predicted a good response. Of the 47 non-responder genes, 13 genes were related to neutrophil granule proteins (NGPs): *AZU1, BPI, CEACAM8, CRISP3, CTSG, DEFA4, ELANE, LCN2, LTF, MMP8, MPO, RNASE2* and *RNASE3*. The combination of three genes: *CMPK2, IFIR1B* and *RNASE3*, showed the greatest predictive power [70]. The expression of neutrophil surface TNF is the another potential marker of anti-TNF responsiveness. TNF is richly expressed on the cell surface of peripheral blood neutrophils of RA patients. However, in anti-TNF responders, the level of its expression decreases in line with disease improvement [71].

## 7. Conclusions

Knowledge regarding anti-TNF pharmacogenetics has not been applied in clinical settings. This is mostly due to the large genetic variability of individuals with RA. The heterogeneity of RA implicates the need for discrimination of different RA subsets with various anti-TNF treatment responses. The other obstacle is that the above-mentioned studies examined only the effects of a single SNP, while identifying the common risk haplotypes may be necessary. Furthermore, the problem with identified specific transcriptional signatures is that their complexity makes them hard to interpret. Applying knowledge of TNF inhibitors’ pharmacogenomics into clinical practice also requires consideration of the importance of environmental factors (e.g., smoking), which are known to affect clinical outcomes. The growing availability of next generation sequencing (NGS) technology and whole genome/exome sequencing (WGS/WES) data, accompanied with statistical tools, can facilitate the identification of rare and low-penetrant causal variants in RA, which is of key importance for the design of genomic tests. Nevertheless, the introducing of specific genomic tests in anti-TNF-treated patients will require prior multicenter validation and standardization.

## Figures and Tables

**Figure 1 ijms-23-02366-f001:**
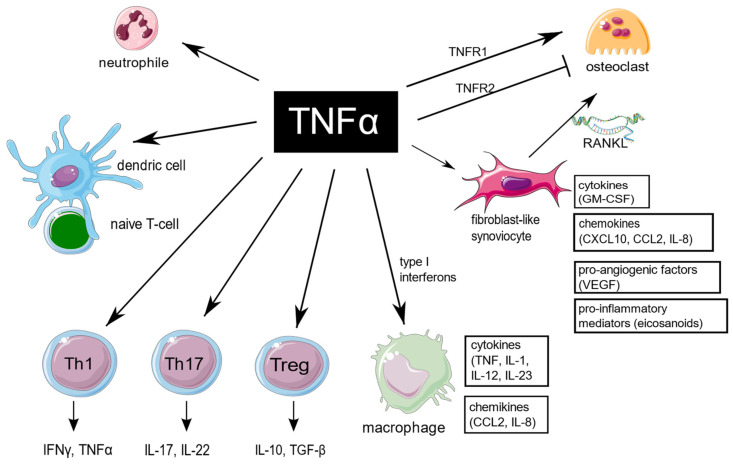
TNF is a pleiotropic cytokine, which is implicated in function of different cells which are important for RA pathogenesis. TNF promote expression of proinflammatory genes in fibroblast-like synoviocytes and macrophages. It has a large impact on T cells by expansion of Th17 lymphocytes subset, transition of Th17 to the non-classic Th1 subset in inflammatory sites, as well as decreasing Tregs levels, and it also enhances leukocyte influx to sites of inflammation. In addition, TNF stimulates neutrophils to delay apoptosis, induce respiratory burst, and upregulate cytokine production. TNF is of key importance in osteoclastogenesis and RA-associated bone loss. The figure was created using the Servier Medical Art template; https://smart.servier.com, accessed on 21 January 2022.

**Figure 2 ijms-23-02366-f002:**
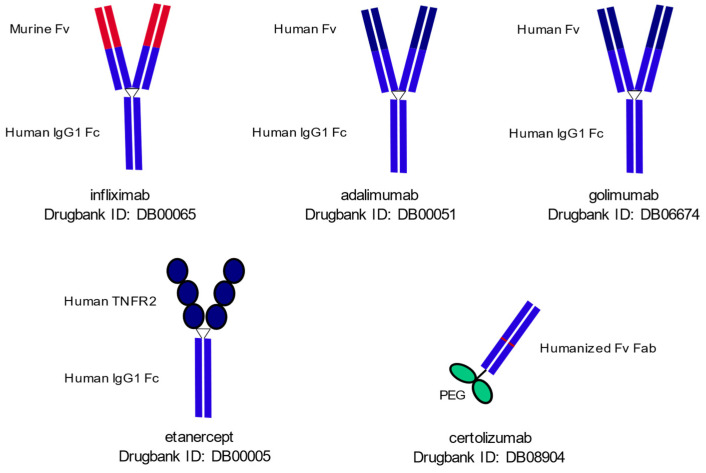
The schematic structure of TNF inhibitors. IFX is comprised of a human IgG1 Fc and murine variable Fab fragment. ADA and GOL are both fully human monoclonal antibodies, containing similar human IgG1 Fc and Fv portions. ETN is a recombinant fusion protein comprised of the extracellular part of the human TNFR2 and human IgG1 Fc. CZP is a Fab’ fragment of a humanized monoclonal antibody, conjugated to PEG. Fab: fragment antigen binding; Fc: fragment crystallizable; Fv: fragment variable; PEG: polyethylene glycol; TNF: tumor necrosis factor alpha.

**Table 1 ijms-23-02366-t001:** Proposed functions of selected polymorphisms, which were identified in GWS or metaanalyses as associated with anti-TNF treatment response in RA.

SNP	Gene	Function	Reference
rs12142623	*PLA2G4A*	Phospholipase enzyme	Umicevic et al.
rs4651370	*PLA2G4A*	Phospholipase enzyme	Umicevic et al.
rs2378945	*NUBPL*	Assembly of the respiratory chain NADH dehydrogenase (complex I)	Umicevic et al.
rs1813443	*NCTN5*	Immunoglobulin superfamily	Umicevic et al.
rs4411591	Unknown	Unclear	Umicevic et al.
rs7767069	Unknown	Unclear	Umicevic et al.
rs1447722	Unknown	Unclear	Umicevic et al.
rs1568885	Unknown	Unclear	Umicevic et al.
rs7767069	*LINC02549*	Unclear	Sánchez-Maldonado et al.
rs717117G	*LARRC55*	Unclear	Sánchez-Maldonado et al.
rs113878252	*MED15*	Polymerase II transcription	Julia et al.
rs6065221	*MAFB*	Regulation of lineage-specific hematopoiesis	Julia et al.
rs6427528	*CD84, Non Coding Transcript Variant LOC105371468*	Self-ligand receptor of the signaling lymphocytic activation molecule	Cui et al.
rs1503860	*CD84: Non Coding Transcript Variant LOC105371468*	Self-ligand receptor of the signaling lymphocytic activation molecule	Cui et al.
rs1800629	*TNF*	Tumor necrosis factor	O’Rielly et al.
rs17301249	*EYA4*	Transcription coactivator and phosphatase	O’Rielly et al.
rs1532269	*PDZD2*	Unclear	Plant et al.
rs2812378	*CCL21*	Chemokine ligand	Farragher et al.

**Table 2 ijms-23-02366-t002:** The analysis of synovial histopathology and synovial gene expression signatures may be helpful to assess probability of response to TNF blockade. The occurrence of different synovial cellular signatures (pathotypes) strongly correlate with specific synovial tissue gene expression and clinical response to TNF and IL-6R inhibitors.

Synovial Pathotype	Gene Expression Profile	Clinical Phenotype
lymphoid	B cell- and plasmablast lineage genes (including *CD19, CD20, cd27, IGLL5, XBP1, immunoglobulin heavy and light chains, CD38 and CXCL13*)	Good clinical response to TNF inhibitors Poor clinical response to IL-6R inhibitors
myeloid	NF-κB pathway genes (including *TNF, IL-1β, IL-1RA, ICAM1,* and *MyD88*), The inflammatory chemokines (*CCL2, IL-8*), Neutrophiles, macrophage and osteoclast—associated genes (*S100A12*, *CD14*, *OSCAR*, respectively)	Poor clinical response to TNF inhibitors Good clinical response to IL-6R inhibitors
fibroid	Fibroblast growth factor genes (*FGF2, FGF9*), Bone homeostasis—associated genes (*BMP6, TNFRSF11b/ osteoprotogerin),* *Wnt* and *TGFβ* pathways genes	Poor clinical response to TNF inhibitors
pauci-immune	*SFRP3/FRZB* gene	Poor clinical response to TNF inhibitors

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
