# Peer review of "Pharmacogenomics of Anti-TNF Treatment Response Marks a New Era of Tailored Rheumatoid Arthritis Therapy"

_ijms, 2022, doi:10.3390/ijms23042366_

Round 1

Reviewer 1 Report

Dear Authors,

Nicely written article,I appreciate your step by step written introduction which brings reader the basic information about RA and TNF. 

My question is what is the next treatment for patients,if they have first/ secondary non response to TNF inhibitors?

Reviewer 2 Report

After carefully reading the manuscript, I conclude that the abstract, introduction, and other chapters cover the issues discussed in an extensive and proper manner.

Although it is a valuable work having an interesting idea it needs some adjustment:

  • As the name of the journal suggests, it is a journal dedicated to molecules. Therefore, it seems necessary to supplement this manuscript with dedicated information about discusses medicines. The authors can provide more information concerning the chemical context of the all discusses anti-TNF drugs. I suggest at least to provide specific ID’s based on the DrugBank or PubChem ID or another. The authors can include the chemical scheme or crystallographic structure of the selected structures.
  • Since a large number of abbreviations are used along the text, I recommend that all used abbreviations should be listed at the end of the main body of text before references. This will certainly provide the reader with a better understanding of the context of the issues discussed. In addition, this is a kind of standard in contemporary scientific papers.
  • The small number of citations of works from the last 5 (after 2018) years is rather surprising (out of 62 bibliographic items, only 15 works have been published after 2018). I believe that the topic has been developing vigorously over the past 5 years and focusing on old articles is not the best approach to reviewing it. Therefore, I am convinced that the authors must carry out a detailed literature review and supplement the citations with references to the latest research in this field.

I recommend publication after minor revision.

Reviewer 3 Report

In this manuscript, the authors reviewed the update of anti-TNF treatment in rheumatoid arthritis (RA) therapy, focusing on pharmacogenomics.  Overall, the report covers the latest studies of the roles of RA at the cellular level and functions. The report was well-organized. However, some minors are required to revise to make it better.

Section 4, characteristics of TNF inhibitors should be expanded to introduce more information for audiences.

In Table 1, rs7767069, remove the comma here.

All the abbreviations across the manuscript should be checked, and the full name for each abbreviation should be listed. For example, line 58 and line 1076 showed twice TNF alpha, etc.

All the gene names should be italic.

Line 251: 3,8%> 3.8%.

Above – mentioned and other words with connection no space, above-mentioned.

cd27 > CD27.
